# A KRAS-directed transcriptional silencing pathway that mediates the CpG island methylator phenotype

Ryan W Serra[1,2†], Minggang Fang[1,2†], Sung Mi Park[1,2], Lloyd Hutchinson[3], Michael R Green[1,2]*

[1]Programs in Gene Function and Expression and Molecular Medicine, University of Massachusetts Medical School, Worcester, United States; [2]Howard Hughes Medical Institute, Chevy Chase, United States; [3]Department of Pathology, University of Massachusetts Medical School, Worcester, United States

**Abstract** Approximately 70% of KRAS-positive colorectal cancers (CRCs) have a CpG island methylator phenotype (CIMP) characterized by aberrant DNA hypermethylation and transcriptional silencing of many genes. The factors involved in, and the mechanistic basis of, CIMP is not understood. Among the CIMP genes are the tumor suppressors $p14^{ARF}$, $p15^{INK4B}$, and $p16^{INK4A}$, encoded by the *INK4-ARF* locus. In this study, we perform an RNA interference screen and identify ZNF304, a zinc-finger DNA-binding protein, as the pivotal factor required for *INK4-ARF* silencing and CIMP in CRCs containing activated KRAS. In KRAS-positive human CRC cell lines and tumors, ZNF304 is bound at the promoters of *INK4-ARF* and other CIMP genes. Promoter-bound ZNF304 recruits a corepressor complex that includes the DNA methyltransferase DNMT1, resulting in DNA hypermethylation and transcriptional silencing. KRAS promotes silencing through upregulation of ZNF304, which drives DNA binding. Finally, we show that ZNF304 also directs transcriptional silencing of *INK4-ARF* in human embryonic stem cells.

**\*For correspondence:** michael. green@umassmed.edu

†These authors contributed equally to this work

**Competing interests:** The authors declare that no competing interests exist.

**Reviewing editor**: Kevin Struhl, Harvard Medical School, United States

## Introduction

Epigenetic dysregulation of gene expression plays a major role in the initiation and progression of cancer (reviewed in *Baylin and Jones, 2011*; *Esteller, 2008*; *Hassler and Egger, 2012*). Among the various epigenetic alterations of cancer genomes, abnormal gains of DNA methylation in normally unmethylated gene promoter CpG islands have been the most extensively investigated. DNA hypermethylation can alter genetic stability and genomic structure, and is associated with transcriptional silencing of gene expression (commonly referred to as epigenetic silencing; reviewed in *Baylin and Jones, 2011*; *Esteller, 2008*; *Hassler and Egger, 2012*). Numerous studies have identified specific genes affecting cellular growth control that become hypermethylated and transcriptionally silenced in many cancers.

Colorectal cancers (CRCs) provide a striking example of alterations in DNA methylation that occur during tumor development. A subset of CRCs have a so-called CpG island methylator phenotype (CIMP) characterized by aberrant DNA hypermethylation of many genes. In fact, CRCs can be categorized into three distinct subclasses based on their epigenetic and genetic profiles: CIMP-1 (also called CIMP-high), characterized by intense methylation of multiple genes, microsatellite instability and BRAF mutations; CIMP-2 (also called CIMP-low), typified by methylation of a more limited group of genes and mutation in KRAS; and CIMP-negative, distinguished by infrequent methylation and p53 mutation (*Yagi et al., 2010*; *Kaneda and Yagi, 2011*). Specific panels of CIMP marker genes have been developed to classify CRCs into these three subclasses. These CIMP marker genes have potential clinical value, and

**eLife digest** Colorectal cancer, which affects the large intestine, is a leading cause of cancer deaths worldwide, ranking fourth after cancers of the lung, stomach, and liver. Like these other cancers, this disease is caused by mutations to genes that allow cells to multiply in an out of control manner. Mutations that change the gene encoding a protein called KRAS are found in many different types of cancer. Moreover, about 70% of colorectal cancers with a KRAS mutation also have an excess of small chemical marks on other genes, some of which are known to suppress the growth of tumors. These marks 'switch off' these genes, and although the identities of the enzymes that typically leave these marks on DNA are known, the link between these enzymes and the KRAS protein is unknown.

Now Serra, Fang et al. have identified a protein, called ZNF304, that is required by KRAS to switch off a large number of genes, including multiple tumor suppressors. In the absence of ZNF304, these tumor suppressor genes remained switched on in cancer cells with the KRAS mutation, so the growth of the tumor was slowed down. ZNF304 is a protein that binds to stretches of DNA, including regions of DNA at the start of several tumor suppressor genes, and it recruits the enzymes that add the chemical marks that switch off these genes.

Serra, Fang et al. found that the levels of ZNF304 protein were elevated in colorectal cancer cells with the mutated KRAS, and showed that this was due to the combined activities of two other proteins that prevented ZNF304 from being broken down in the cell. Mutant KRAS caused an increase in the levels of these two proteins, which in turn caused the elevated ZNF304 levels and the excessive marking of the DNA in the tumor suppressor genes.

Furthermore, some of these same tumor suppressor genes are switched off in the earliest cells in a human embryo—which have the potential to become any of 200 or so cell types in the human body. In these embryonic stem cells, Serra, Fang et al. showed that ZNF304, but not KRAS, was also involved in keeping these genes switched off until the stem cells started changing into specific types of cells.

Since they are a crucial part of the pathway linking a cancer-causing mutation to increased tumor growth, the proteins identified by Serra, Fang et al. could represent promising targets for the development of new anti-cancer drugs.

are currently being evaluated as biomarkers for risk, diagnosis, prognosis, and prediction of therapeutic responsiveness (reviewed in *Egger et al., 2012*; *Gyparaki et al., 2013*).

Among the CIMP genes are three well-known tumor suppressors: $p14^{ARF}$, $p16^{INK4A}$ (also known as *CDKN2A*), and $p15^{INK4B}$ (also known as *CDKN2B*) (reviewed in *Gil and Peters, 2006*). These three genes are located in close proximity to one another (within a 35 kb region) at the *INK4-ARF* locus, yet each is transcribed from a distinct promoter. Interestingly, $p14^{ARF}$ and $p16^{INK4A}$ share exons two and three, but each is translated in a different reading frame, yielding unrelated polypeptides. Inactivation of the *INK4-ARF* locus is one of the most frequent events in cancers (reviewed in *Kim and Sharpless, 2006*). For example, *INK4-ARF* is transcriptionally silenced in 30–45% of all CRCs and in 70% of CRCs that harbor an activating KRAS mutation (*Burri et al., 2001*; *Dominguez et al., 2003*; *Lind et al., 2004*).

The *INK4-ARF* locus is also silenced in some non-malignant cells. For example, *INK4-ARF* is silenced in embryonic, fetal, and adult stem cells, but in more differentiated cells, it becomes poised for expression and increasingly responsive to aberrant mitogenic signals such as those elicited by activated oncogenes (reviewed in *Sherr, 2012*). This process is reversed when somatic cells are induced to regain pluripotency through reprogramming. Expression of *INK4-ARF* limits stem cell self-renewal, suggesting that coordinated *INK4-ARF* expression may normally act to restrict stem cell numbers. Accordingly, the *INK4-ARF* locus has been shown to be a barrier for reprogramming (*Li et al., 2009*).

In actively growing human diploid fibroblasts, the *INK4A-ARF* locus is silenced by histone H3 lysine 27 trimethylation (H3K27me3) directed by Polycomb group proteins. When such cells are exposed to cellular stress, such as oncogenic signals, the H3K27me3 mark on the locus is decreased, resulting in expression of *INK4A-ARF* genes (*Jacobs et al., 1999*; *Bracken et al., 2007*; *Kotake et al., 2007*). Transcriptional activation is due, at least in part, to upregulation of the H3K27 demethylase JMJD3,

which removes H3K27me3 from *INK4A-ARF* (*Agger et al., 2009*). Whether the mechanism of *INK4-ARF* silencing in stem cells and primary differentiated cells is related to, or distinct from, that in cancer cells is unknown.

The factors, regulatory pathways, and mechanisms underlying the aberrant promoter hypermethylation and transcriptional silencing characteristic of CIMP-positive CRCs remain to be determined. In addition, the relationship between the initiating genetic events responsible for tumorigenesis (e.g., acquisition of activating mutations in oncogenes) and the epigenetic alterations in CIMP-positive CRCs is not understood. To begin to address these questions, in this study, using *p14^{ARF}* as a representative CIMP gene, we perform an RNA interference (RNAi) screen to identify factors required for *p14^{ARF}* silencing. Our results reveal a KRAS-directed pathway that mediates silencing of the entire *INK4-ARF* locus, is responsible for CIMP in CRCs, and is related to the pathway that silences *INK4-ARF* in human embryonic stem cells (hESCs).

## Results

### An RNAi screen to identify mediators of *INK4-ARF* transcriptional silencing

To screen for factors involved in transcriptional silencing of *INK4-ARF*, we generated a reporter construct in which the *p14^{ARF}* promoter was used to direct expression of the blasticidin-resistance (*Blast^R*) gene (*Figure 1A*). This *p14^{ARF}-Blast^R* reporter construct was stably transduced into DLD-1 cells, a human CRC cell line in which endogenous *p14^{ARF}* is transcriptionally silenced (*Zheng et al., 2000*; *Figure 1B*). We selected cells in which the reporter gene had been silenced, as evidenced by acquisition of blasticidin resistance (*Figure 1C*), transcriptional derepression (*Figure 1B*), and decreased DNA hypermethylation (*Figure 1D*) following treatment with the DNA methyltransferase inhibitor 5-aza-2′-deoxycytidine.

A genome-wide human small hairpin (shRNA) library (*Silva et al., 2005*) comprising ~62,400 shRNAs was divided into 10 pools, which were packaged into retrovirus particles and used to stably transduce the DLD-1/*p14^{ARF}-Blast^R* reporter cell line. Blasticidin-resistant colonies, indicative of derepression of the reporter gene, were selected and the shRNAs identified by sequence analysis (*Figure 1A*).

Positive candidates identified in the primary screen were validated by stably transducing DLD-1 cells with an shRNA directed against each candidate gene, followed by the analysis of endogenous *p14^{ARF}* expression by quantitative RT-PCR (qRT-PCR). Using this approach, we identified eight genes that, following shRNA-mediated knockdown, resulted in derepression of endogenous *p14^{ARF}* (*Figure 2A*, *Figure 2—figure supplement 1*). qRT-PCR analysis confirmed that each shRNA reduced target gene expression (*Figure 2—figure supplement 2*). For all genes, a second shRNA whose sequence was unrelated to that isolated from the primary screen also resulted in target gene knockdown (*Figure 2—figure supplement 3A*) and derepression of endogenous *p14^{ARF}* (*Figure 2—figure supplement 3B*).

### Identification of a ZNF304-corepressor complex required for transcriptional silencing of *INK4-ARF*

Our previous studies have shown that transcriptional silencing of tumor suppressor genes (TSGs) involves a sequence-specific DNA-binding protein (*Gazin et al., 2007*; *Palakurthy et al., 2009*). Therefore, we elected to focus on ZNF304, a zinc finger DNA-binding protein that contains a KRAB repressor domain (*Sabater et al., 2002*). The qRT-PCR and immunoblot results of *Figure 2A,B* show that in addition to *p14^{ARF}*, knockdown of ZNF304 derepressed *p15^{INK4B}* and *p16^{INK4A}*, which are also transcriptionally silenced in DLD-1 cells (*Zheng et al., 2000*; *Ishiguro et al., 2006*).

KRAB domain proteins function by recruiting a corepressor complex that includes the scaffolding protein KAP1 and the histone methyltransferase SETDB1 (*Iyengar and Farnham, 2011*). Knockdown of KAP1 or SETDB1 (*Figure 2—figure supplement 4A*) resulted in derepression of *p14^{ARF}*, *p15^{INK4B}*, and *p16^{INK4A}* (*Figure 2C*). Similar results were obtained using a second, unrelated KAP1 or SETDB1 shRNA (*Figure 2—figure supplement 4B,C*). *p14^{ARF}*, *p15^{INK4B}*, and *p16^{INK4A}* were also substantially derepressed following knockdown of DNMT1 but not the other DNA methyltransferases, DNMT3A, or DNMT3B (*Figure 2C*, *Figure 2—figure supplement 4*). The identification of KAP1, SETDB1, and DNMT1 as ZNF304 corepressors indicates that our shRNA screen, like other large-scale shRNA screens (reviewed in *Mullenders and Bernards, 2009*), was not saturating.

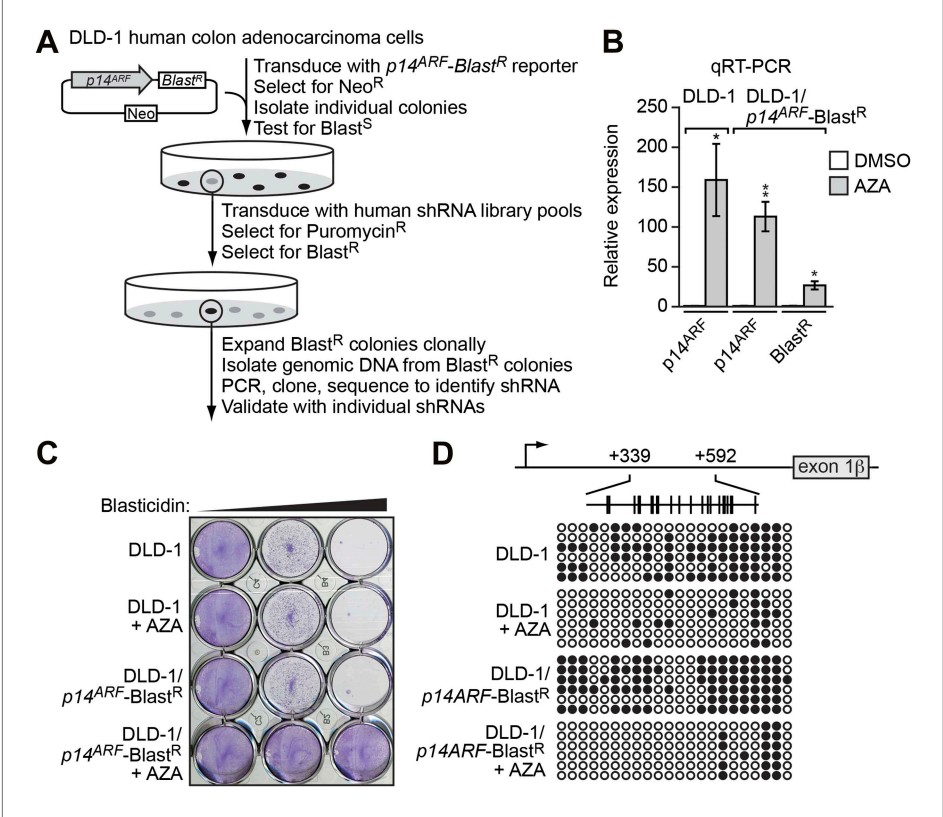

**Figure 1**. Derivation and validation of the DLD-1/*p14ARF-BlastR* reporter cell line. (**A**) Schematic of the shRNA screen. (**B**) qRT-PCR analysis monitoring *p14ARF* expression in parental DLD-1 cells, or *p14ARF* and *BlastR* expression in DLD-1/*p14ARF*-BlastR cells, following treatment with either DMSO or 5-aza-2'-deoxycytidine (AZA). The results were normalized to that observed upon DMSO treatment, which was set to 1. Data are represented as mean ± SD. *p≤0.05, **p≤0.01. (**C**) Viability of DLD-1 or DLD-1/*p14ARF-BlastR* cells treated with DMSO or AZA for 3 days and then 0, 5, or 10 μM blasticidin for 6 days. Cells were stained with crystal violet. (**D**) Bisulfite sequencing analysis of the endogenous *p14ARF* promoter in parental DLD-1 cells or the *p14ARF-BlastR* reporter in DLD-1/*p14ARF-BlastR* cells treated in the absence or presence of AZA. (Top) Schematic of the *p14ARF* promoter; positions of CpGs are shown to scale by vertical lines. (Bottom) Each circle represents a methylated (black) or unmethylated (white) CpG dinucleotide. Each row represents a single clone.

The chromatin immunoprecipitation (ChIP) assay of **Figure 2D** shows that ZNF304, as well as KAP1, SETDB1 and DNMT1, were bound to the *p14ARF*, *p15INK4B*, and *p16INK4A* promoters in DLD-1 cells. Notably, knockdown of ZNF304 substantially decreased binding of KAP1, SETDB1, and DNMT1. Knockdown of ZNF304, KAP1, SETDB1 or DNMT1 also decreased *p14ARF* promoter hypermethylation (**Figure 2E**).

We predicted that the loss of ZNF304, which results in derepression of the *INK4-ARF* locus, would reduce tumorigenicity. Consistent with this prediction, **Figure 2F** shows that shRNA-mediated knockdown of ZNF304 in DLD-1 cells significantly suppressed tumor growth in mouse xenografts. shRNA-mediated knockdown of DNMT1 similarly suppressed tumor growth, consistent with previous results (**Morita et al., 2013**).

## Activated KRAS-mediated upregulation of ZNF304 is required for transcriptional silencing of *INK4-ARF*

DLD-1 cells contain an activated KRAS(G13D) mutation and we therefore investigated the relationship between KRAS and silencing of *INK4-ARF.* shRNA-mediated knockdown of KRAS in DLD-1 cells (**Figure 3—figure supplement 1A,B**) resulted in derepression of *p14ARF*, *p15INK4A*, and *p16INK4B* (**Figure 3A**, **Figure 3—figure supplement 1C**) and substantially reduced binding of ZNF304 and its corepressors to all three promoters (**Figure 3B**). Similarly, treatment with manumycin A, a RAS farnesyltransferase inhibitor (**Hara et al., 1993**), also resulted in derepression of *p14ARF*, *p15INK4B*, and *p16INK4A* (**Figure 3C**)

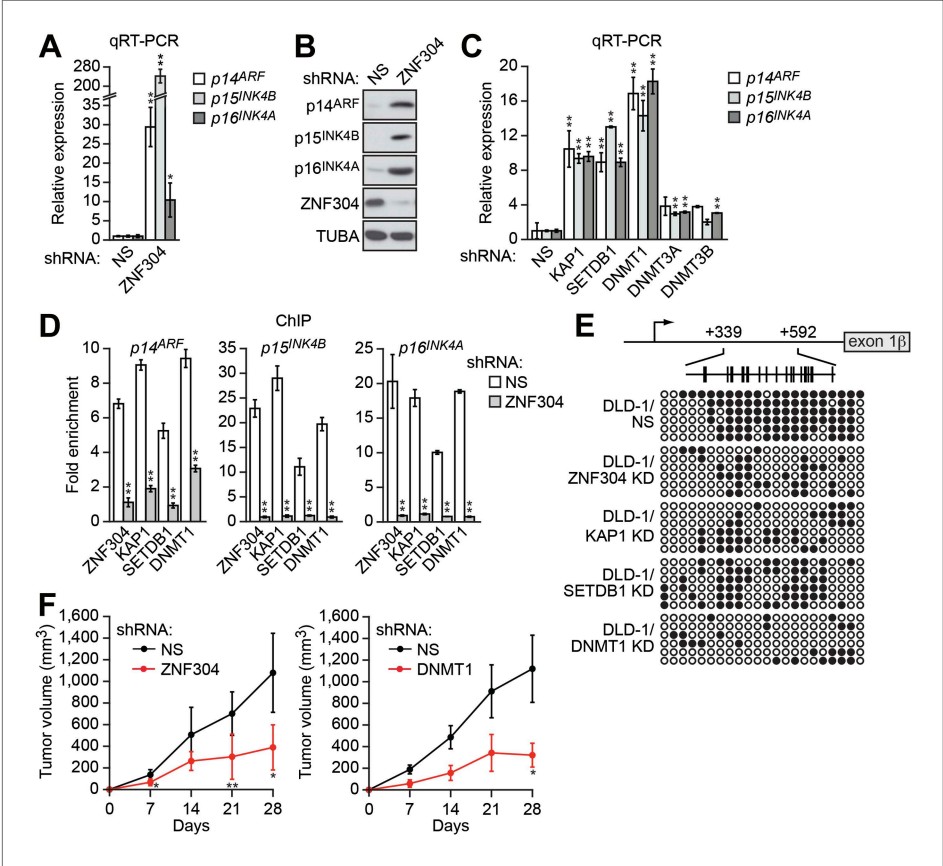

**Figure 2**. Identification of a ZNF304-corepressor complex required for transcriptional silencing of *INK4-ARF* in CRCs. (**A**) qRT-PCR analysis monitoring *INK4-ARF* expression in DLD-1 cells expressing a non-silencing (NS) or ZNF304 shRNA. The results were normalized to that obtained with the NS control, which was set to 1. (**B**) Immunoblot analysis monitoring INK4-ARF levels in DLD-1 cells expressing a NS or ZNF304 shRNA. α-tubulin (TUBA) was monitored as a loading control. (**C**) qRT-PCR analysis monitoring *INK4-ARF* expression in DLD-1 cells expressing a NS, KAP1, SETDB1, DNMT1, DNMT3A, or DNMT3B shRNA. (**D**) ChIP assay monitoring binding of ZNF304, KAP1, SETDB1 and DNMT1 to *INK4-ARF* promoters in DLD-1 cells expressing a NS or ZNF304 shRNA. The results were normalized to that obtained with IgG, which was set to 1. (**E**) Bisulfite sequencing analysis of the *p14^ARF* promoter in DLD-1 cells expressing a NS, KAP1, SETDB1, or DNMT1 shRNA. (**F**) Tumor formation assay. DLD-1 cells expressing a NS and ZNF304 (left) or DNMT1 (right) shRNA were subcutaneously injected into the flanks of nude mice (n = 3), and tumor formation was measured. Data are represented as mean ± SD. *p≤0.05, **p≤0.01. Results from experiments showing validation of candidates from the RNAi screen, and ZNF304 corepressors, for a role in *INK4-ARF* transcriptional silencing in DLD-1 cells are presented in ***Figure 2—figure supplements 1 and 2***.

The following figure supplements are available for figure 2:

**Figure supplement 1**. Validation of candidates from the RNAi screen for a role in *INK4-ARF* transcriptional silencing in DLD-1 cells.

**Figure supplement 2**. Knockdown efficiencies of candidate shRNAs isolated from the RNAi screen.

**Figure supplement 3**. Validation of candidates from the RNAi screen for a role in *p14^ARF* transcriptional silencing in DLD-1 cells using a second shRNA.

**Figure supplement 4**. Validation of ZNF304 corepressors for a role in *INK4-ARF* transcriptional silencing in DLD-1 cells.

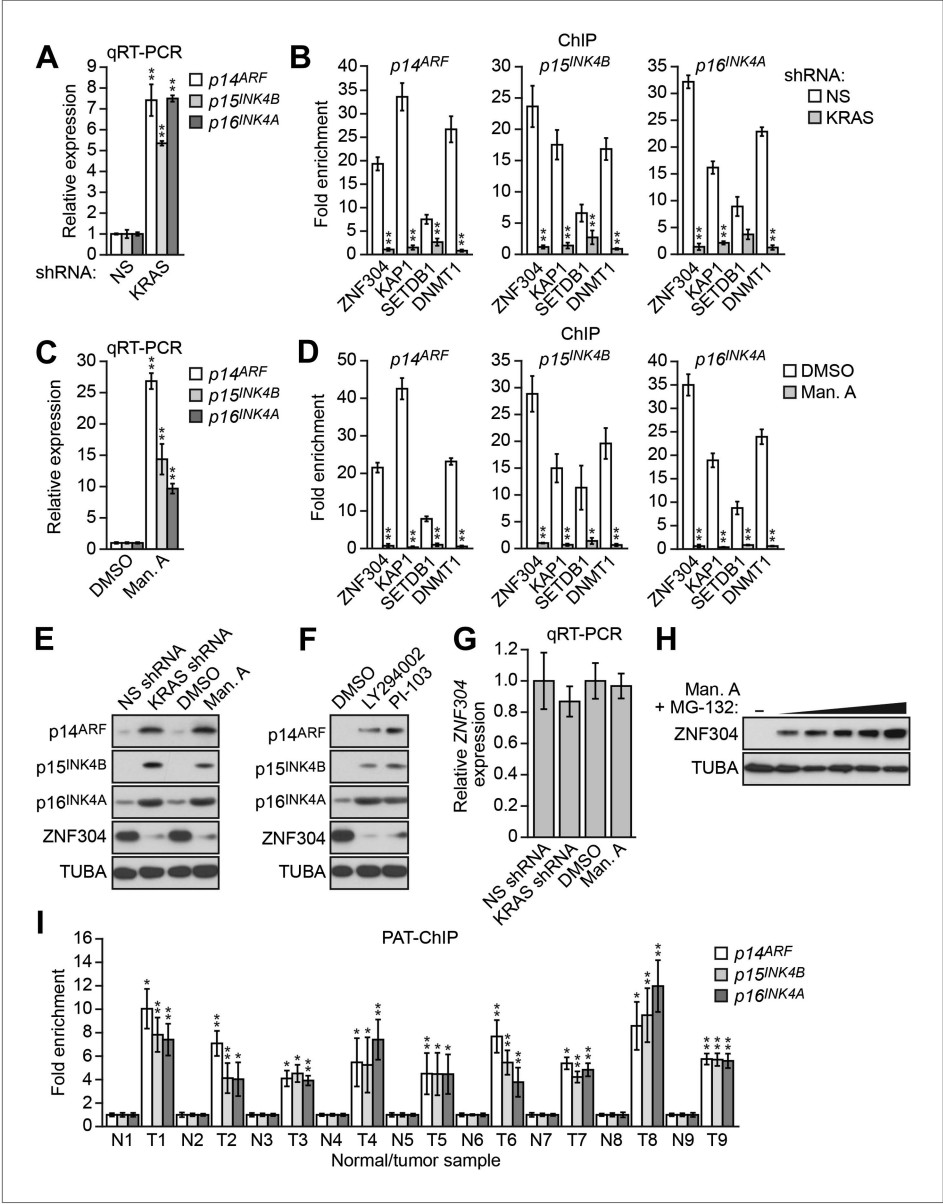

**Figure 3**. Activated KRAS-mediated upregulation of ZNF304 is required for transcriptional silencing of *INK4-ARF*. (**A**) qRT-PCR analysis monitoring *INK4A-ARF* expression in DLD-1 cells expressing a NS or KRAS shRNA. (**B**) ChIP analysis monitoring binding of ZNF304, KAP1, SETDB1, and DNMT1 to *INK4-ARF* promoters in DLD-1 cells expressing a NS or KRAS shRNA. (**C**) qRT-PCR analysis monitoring *INK4A-ARF* expression in DLD-1 cells treated with DMSO or manumycin A (Man. A). The results were normalized to DMSO, which was set to 1. (**D**) ChIP analysis monitoring binding of ZNF304, KAP1, SETDB1, and DNMT1 to *INK4-ARF* promoters in DLD-1 cells treated with DMSO or Man. A. (**E**) Immunoblot analysis showing INK4-ARF levels in DLD-1 cells treated with a NS or KRAS shRNA, or DMSO or Man. A. (**F**) Immunoblot analysis showing INK4-ARF levels in DLD-1 cells treated with DMSO, LY294002, or PI-103. (**G**) qRT-PCR analysis monitoring *ZNF304* expression in DLD-1 cells treated with a NS or KRAS shRNA, or DMSO or Man. A. (**H**) Immunoblot analysis showing ZNF304 levels in DLD-1 cells treated with Man. A for 24 hr and 0–10 μM MG-132 for 4 hr. (**I**) PAT-ChIP analysis monitoring binding of ZNF304 to *INK4-ARF* promoters in matched adjacent normal (N) and KRAS-positive CRC human tumor (T) samples. Results were normalized to normal samples, which were set to 1. Data are represented as mean ± SD. *p≤0.05, **p≤0.01. Results from experiments validating KRAS knockdown efficiency and the role of KRAS in repressing *p14^ARF* expression, as well as experiments validating the role of ZNF304 and its corepressors in *INK4-ARF* silencing in other KRAS-positive CRC cell lines, are presented in *Figure 3—figure supplements 1–4*.

*Figure 3. Continued on next page*

*Figure 3. Continued*

The following figure supplements are available for figure 3:

**Figure supplement 1**. Validation of a role for KRAS in *p14^ARF* transcriptional silencing in DLD-1 cells.

**Figure supplement 2**. ZNF304 and its corepressors bind to the *INK4-ARF* promoters in other KRAS-positive human CRC cell lines.

**Figure supplement 3**. Validation of a role for ZNF304 and KRAS in *INK4-ARF* transcriptional silencing in other KRAS-positive human CRC cell lines.

**Figure supplement 4**. The *p14^ARF* promoter is hypermethylated in KRAS-positive human CRC samples.

and reduced binding of ZNF304 and its corepressors to the three promoters (**Figure 3D**). Notably, shRNA-mediated knockdown or pharmacological inhibition of KRAS markedly reduced ZNF304 protein levels (**Figure 3E**). Likewise, addition of the phosphoinositide 3-kinase (PI3K) inhibitor LY294002 (**Vlahos et al., 1994**) or PI-103, which blocks the PI3K-AKT signaling pathway downstream of activated KRAS (**Hayakawa et al., 2006**), also derepressed p14^ARF, p15^INK4B, and p16^INK4A and reduced ZNF304 levels (**Figure 3F**). By contrast, following shRNA-mediated knockdown or pharmacological inhibition of KRAS, *ZNF304* mRNA levels were not significantly affected (**Figure 3G**). Thus, upregulation of ZNF304 by activated KRAS is predominantly post-transcriptional. Consistent with this idea, the reduction of ZNF304 protein levels following KRAS inhibition could be counteracted by proteasome inhibition (**Figure 3H**).

## Validation of the role of ZNF304 in *INK4-ARF* silencing in other KRAS-positive human CRC cell lines and tumor samples

To determine the generality and clinical relevance of these results, we analyzed other KRAS-positive human CRC cell lines and tumor samples. In HCT116 and HCT15 CRC cell lines, in which *INK4-ARF* is transcriptionally silenced (**Yagi et al., 2010**), ZNF304 and its corepressors were associated with the p14^ARF, p15^INK4B, and p16^INK4A promoters (**Figure 3—figure supplement 2**). Moreover, shRNA-mediated knockdown of ZNF304 or KRAS (**Figure 3—figure supplement 3A,B**) or treatment with manumycin A (**Figure 3—figure supplement 3C**) derepressed p14^ARF, p15^INK4B, and p16^INK4A expression.

We next used a pathology tissue ChIP (PAT-ChIP) assay (**Fanelli et al., 2011**) to measure association of ZNF304 with p14^ARF, p15^INK4B, and p16^INK4A promoters in KRAS-positive human CRC tumor samples and, as a control, adjacent matched normal colon. **Figure 3I** shows that ZNF304 was substantially enriched at the three promoters in CRC tumor samples relative to adjacent normal colon. Bisulfite sequencing analysis confirmed *p14^ARF* promoter hypermethylation in the KRAS-positive CRC tumors but not in the matched normal colon (**Figure 3—figure supplement 4**).

## Activated KRAS upregulates ZNF304 through the deubiquitinase USP28

Another factor isolated in our primary RNAi screen was USP28 (**Figure 2—figure supplement 1**), a nuclear-localized deubiquitinase (**Popov et al., 2007**). We therefore asked whether USP28 was responsible for stabilization of ZNF304. Similar to the results with KRAS, knockdown of USP28 in DLD-1 cells substantially reduced ZNF304 protein (**Figure 4A**) but not mRNA (**Figure 4B**) levels. The co-immunoprecipitation experiment of **Figure 4C** shows that USP28 and ZNF304 were physically associated. Moreover, co-transfection of DLD-1 cells with wild-type USP28, but not a catalytically inactive USP28(C171A) mutant (**Popov et al., 2007**), reduced ubiquitination of ZNF304 (**Figure 4D**). Notably, shRNA-mediated knockdown or pharmacological inhibition of KRAS led to reduced levels of USP28 protein (**Figure 4E**) and mRNA (**Figure 4F**).

The results of **Figure 4F** showed that KRAS promoted the transcriptional upregulation of *USP28*. To gain insight into the basis of KRAS-mediated regulation of *USP28* transcription, we performed bioinformatic analysis on the *USP28* promoter and identified two putative binding sites for the transcription factor cJUN (**Figure 4—figure supplement 1**). Notably, previous studies have shown that KRAS increases cJUN activity (**Mechta et al., 1997**). The ChIP results of **Figure 4G** show, consistent with the bioinformatic prediction, that in DLD-1 cells cJUN was bound to the *USP28* promoter. Moreover, knockdown of cJUN (**Figure 4—figure supplement 2A**) resulted in decreased expression of *USP28*

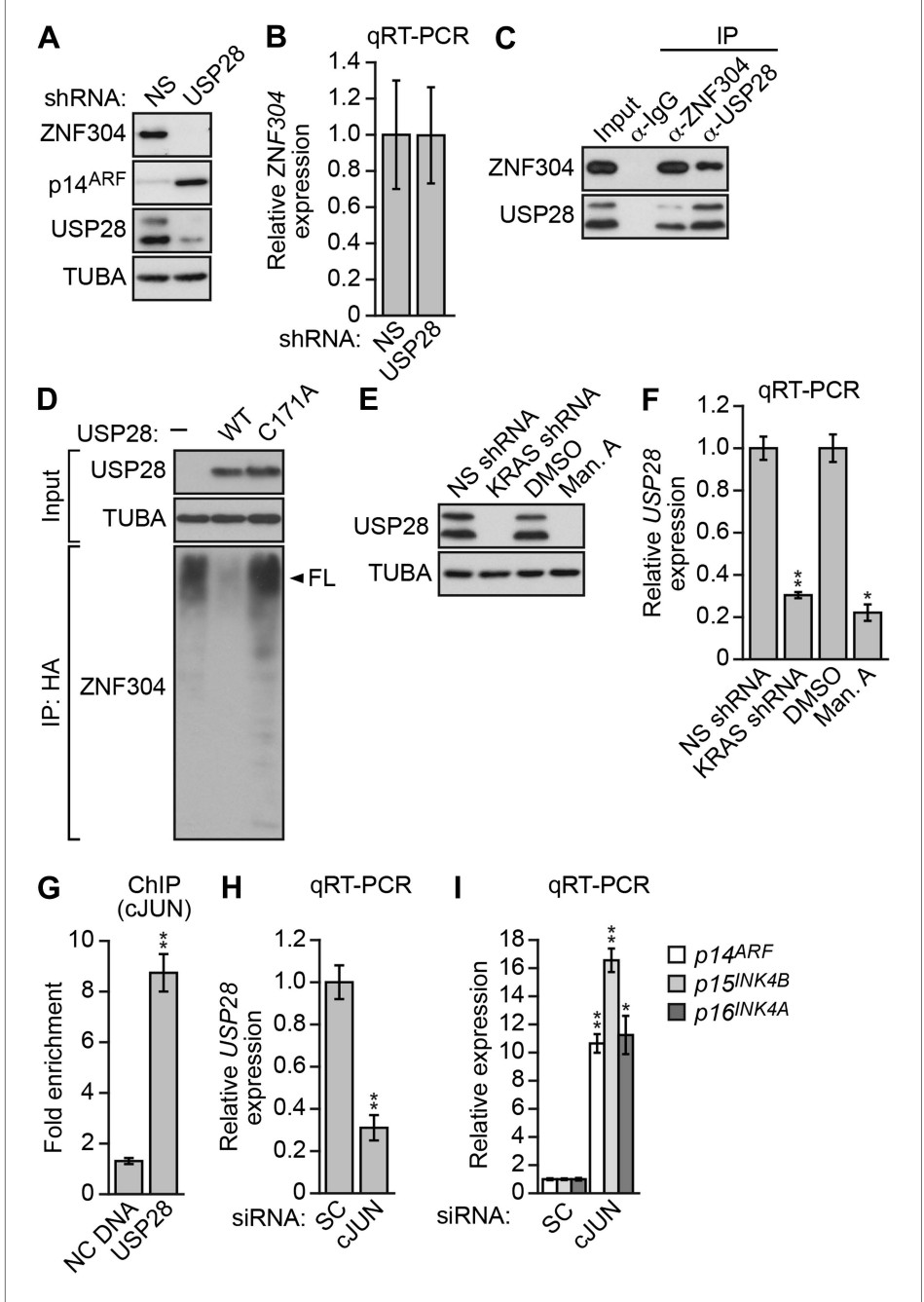

**Figure 4**. Activated KRAS upregulates ZNF304 through the deubiquitinase USP28. (**A**) Immunoblot analysis showing ZNF304 levels in DLD-1 cells expressing a NS or USP28 shRNA. (**B**) qRT-PCR analysis monitoring *ZNF304* expression in DLD-1 cells expressing a NS or USP28 shRNA. (**C**) Co-immunoprecipitation analysis. DLD-1 cell extracts were immunoprecipitated with a ZNF304, USP28 or control (IgG) antibody, and the immunoprecipitate was analyzed for ZNF304 or USP28 by immunoblotting. (**D**) HA-ubiquitination pull-down assay. Extracts from 293T cells expressing HA-tagged ubiquitin, FLAG-tagged ZNF304, and FLAG-tagged wild-type (WT) or mutant (C171A) USP28 were immunoprecipitated using an HA antibody, and the immunoprecipitate was analyzed using a ZNF304 antibody. The arrowhead indicates the position of full-length (FL) ZNF304. (**E**) Immunoblot analysis showing USP28 levels in DLD-1 cells treated with a KRAS shRNA or inhibitor. (**F**) qRT-PCR analysis monitoring *USP28* expression in DLD-1 cells treated with a KRAS shRNA or inhibitor. (**G**) ChIP analysis monitoring binding of cJUN to the *USP28* promoter or an irrelevant negative control (NC) DNA region. (**H** and **I**) qRT-PCR analysis monitoring *USP28* (**H**) or

*Figure 4. Continued on next page*

*Figure 4. Continued*

*INK4-ARF* (**I**) expression in DLD-1 cells expressing a control scrambled (SC) or cJUN siRNA. Experiments validating the role of cJUN in regulating *USP28* expression are presented in *Figure 4—figure supplements 1,2*.
The following figure supplements are available for figure 4:
**Figure supplement 1**. The *USP28* promoter contains consensus cJUN-binding sites.
**Figure supplement 2**. cJUN transcriptionally stimulates *USP28* expression in KRAS-positive DLD-1 cells.

(*Figure 4H*, *Figure 4—figure supplement 2B*) and derepression of $p14^{ARF}$, $p15^{INK4B}$, and $p16^{INK4A}$ (*Figure 4I*, *Figure 4—figure supplement 2C*). Thus, cJUN is responsible, at least in part, for RAS-mediated transcriptional upregulation of *USP28*.

## Role of the protein kinase PRKD1 in KRAS-mediated stabilization of ZNF304

Previous studies have shown that deubiquitinase–substrate interactions are often regulated by phosphorylation (*Kessler and Edelmann, 2011*). Notably, USP28 contains two predicted phosphorylation sites for PRKD1 (consensus sequence LxRxxS, [*Nishikawa et al., 1997*]; *Figure 5A*), a serine/threonine protein kinase (*Johannes et al., 1994*) isolated in our RNAi screen (*Figure 2—figure supplement 1*) that is dysregulated in a variety of cancers (*Sundram et al., 2011*). We therefore analyzed the role of PRKD1 in KRAS-mediated stabilization of ZNF304. shRNA-mediated knockdown of PRKD1 in DLD-1 cells resulted in decreased ZNF304 protein levels (*Figure 5B*), whereas *ZNF304* mRNA levels were not significantly affected (*Figure 5C*). Furthermore, treatment of DLD-1 cells with a PRKD1 chemical inhibitor, CRT0066101 (*Harikumar et al., 2010*), also resulted in decreased ZNF304 protein levels (*Figure 5D*), derepression of $p14^{ARF}$, $p15^{INK4B}$, and $p16^{INK4A}$ (*Figure 5E*), and loss of ZNF304, KAP1, SETDB1, and DNMT1 binding to the three promoters (*Figure 5F*). Similar to the results with USP28, PRKD1 protein and mRNA levels decreased following shRNA-mediated knockdown or pharmacological inhibition of KRAS (*Figure 5G,H*).

We next asked whether USP28 is a substrate of PRKD1. Consistent with this possibility, we found that USP28 and PRKD1 were stably associated in a co-immunoprecipitation assay (*Figure 5I*). We then performed an in vitro kinase assay with purified PRKD1 whose activity was verified in an auto-phosphorylation assay (*Figure 5—figure supplement 1*). *Figure 5J* shows that PRKD1 could phosphorylate a USP28 peptide containing the second, more conserved predicted phosphorylation site (*Figure 5A*). Notably, unlike wild-type USP28, a USP28 derivative containing a mutation in the PRKD1 phosphorylation site, USP28(S899A), was unable to reduce ZNF304 ubiquitination (*Figure 5K*). Likewise, treatment with the PRKD1 inhibitor CRT0066101 prevented USP28 from reducing ubiquitination of ZNF304 (*Figure 5—figure supplement 2*).

The results of *Figure 5H* demonstrated that KRAS promoted the transcriptional upregulation of *PRKD1*. To gain mechanistic insight into this process, we performed bioinformatic analysis on the *PRKD1* promoter and identified consensus binding sites for the intestine-specific transcription factor CDX1 (*Figure 5—figure supplement 3*). Notably, previous studies have shown that KRAS positively regulates CDX1 activity through the mitogen-activated protein kinase pathway (*Lorentz et al., 1999*). The ChIP results of *Figure 5L* show, consistent with the bioinformatic prediction, that CDX1 was bound to the *PRKD1* promoter in DLD-1 cells. Moreover, knockdown of CDX1 (*Figure 5–figure supplement 4A*) resulted in decreased expression of *PRKD1* (*Figure 5M*, *Figure 5—figure supplement 4B*) and derepression of $p14^{ARF}$, $p15^{INK4B}$, and $p16^{INK4A}$ (*Figure 5N*, *Figure 5—figure supplement 4C*) Thus, CDX1 is responsible, at least in part, for RAS-mediated transcriptional upregulation of *PRKD1*.

## The ZNF304-corepressor complex mediates CIMP in KRAS-positive CRCs

As stated above, approximately 70% of CRCs containing activated KRAS are CIMP-positive as defined by aberrant hypermethylation of a representative panel of ~50 CIMP marker genes (*Yagi et al., 2010*; *Kaneda and Yagi, 2011*). Notably, these CIMP marker genes include $p14^{ARF}$ and $p16^{INK4A}$, as well as other TSGs, which prompted us to ask whether ZNF304 and its corepressors have a general role in the aberrant hypermethylation and transcriptional silencing characteristic of CIMP-positive CRCs containing activated KRAS. To address this possibility, we knocked down KRAS or ZNF304 in DLD-1 cells and analyzed

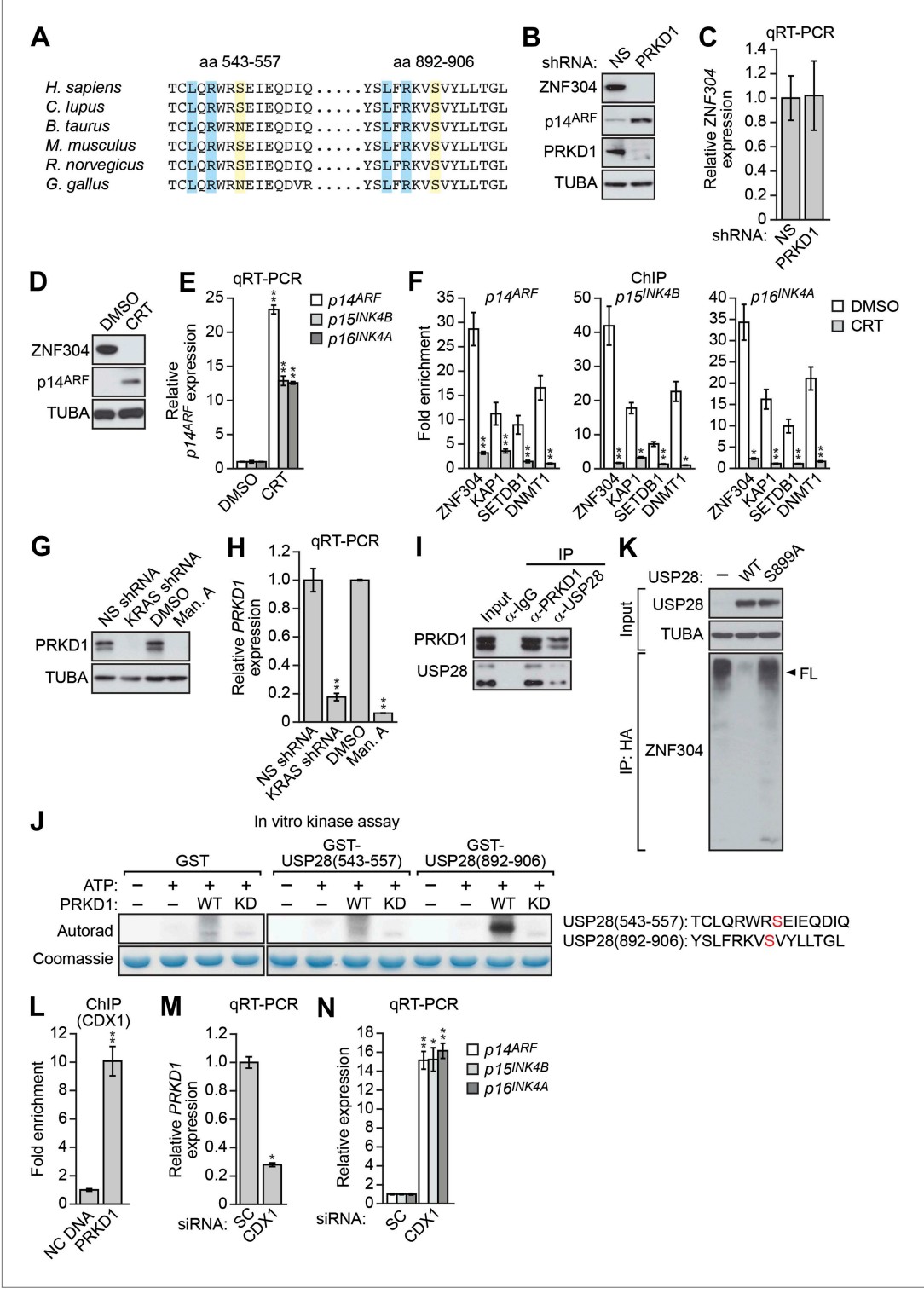

**Figure 5**. Role of the protein kinase PRKD1 in KRAS-mediated stabilization of ZNF304. (**A**) Multiple sequence alignment of the two putative PRKD1 phosphorylation sites in USP28. Blue indicates conserved leucine and arginine residues in the PRKD1 phosphorylation consensus sequence, and yellow indicates the putative phosphorylated serine residue. The alignment was performed using NCBI's HomoloGene; amino acid numbers refer to the human protein. (**B** and **C**) Immunoblot (**B**) and qRT-PCR (**C**) analysis monitoring ZNF304 in DLD-1 cells expressing a NS or PRKD1 shRNA. (**D**) Immunoblot analysis showing p14ARF levels in DLD-1 cells treated with DMSO or CRT0066101 (CRT). (**E**) qRT-PCR

*Figure 5. Continued on next page*

*Figure 5. Continued*

analysis monitoring *INK4-ARF* expression in DLD-1 cells treated with DMSO or CRT. (**F**) ChIP monitoring ZNF304 and corepressor binding to *INK4-ARF* in CRT0066101-treated DLD-1 cells. (**G** and **H**) PRKD1 immunoblot (**G**) and qRT-PCR (**H**) in DLD-1 cells treated with a KRAS shRNA or inhibitor. (**I**) Co-immunoprecipitation analysis. DLD-1 cell extracts were immunoprecipitated with a PRKD1, USP28 or control (IgG) antibody, and the immunoprecipitate was analyzed for PRKD1 or USP28 by immunoblotting. (**J**) (Left) In vitro kinase assay. Purified wild-type (WT) or kinase dead (KD) PRKD1 was incubated with USP28 peptides (shown on the right) and γ-ATP and analyzed for incorporation of the radiolabel by autoradiography. (**K**) HA-ubiquitination pull-down assay as described in *Figure 4D* except 293T cells expressed WT or mutant (S899A) USP28. (**L**) ChIP analysis monitoring binding of CDX1 to the *PRKD1* promoter or an irrelevant DNA region (NC) DNA. (**M** and **N**) qRT-PCR analysis monitoring *PRKD1* (**M**) or *INK4-ARF* (**N**) expression in DLD-1 cells expressing a control scrambled (SC) or CDX1 siRNA. Data are represented as mean ± SD. *p≤0.05, **p≤0.01. Control experiments related to *Figure 5* are presented in *Figure 5—figure supplements 1–4*.

The following figure supplements are available for figure 5:

**Figure supplement 1**. Confirmation of in vitro autophosphorylation activity of wild-type PRKD1 but not a kinase-dead PRKD1 mutant.

**Figure supplement 2**. USP28-mediated deubiquitination of ZNF304 does not occur in the presence of the PRKD1 chemical inhibitor CRT0066101.

**Figure supplement 3**. The *PRKD1* promoter contains consensus CDX1-binding sites.

**Figure supplement 4**. CDX1 transcriptionally stimulates *PRKD1* expression in KRAS-positive DLD-1 cells.

CIMP marker gene expression by qRT-PCR. Remarkably, knockdown of either KRAS or ZNF304 derepressed expression of all 50 CIMP marker genes analyzed (*Figure 6A*). Interestingly, knockdown of either KRAS or ZNF304 also derepressed *VIM* and *SEPT9* (*Figure 6—figure supplement 1*), whose DNA hypermethylation is used to diagnose CRC (*Gyparaki et al., 2013*). Bisulfite sequencing analysis of a representative subset of CIMP marker genes, which included *p14^ARF*, *p16^INK4A* and seven additional genes, revealed that shRNA-mediated knockdown of KRAS or ZNF304 also decreased promoter hypermethylation (*Figure 2E*, *Figure 6—figure supplement 2*). ChIP analysis showed significant enrichment of ZNF304, KAP1, SETDB1, and DNMT1 on the promoters of the nine CIMP marker genes (*Figures 2D and 6B*), whose expression was also derepressed by the knockdown of the ZNF304 corepressors (*Figures 2C and 6C*).

We next asked whether ZNF304 and its corepressors silenced CIMP marker genes in other KRAS-positive human CRC cell lines and tumor samples. In CIMP-positive HCT116 and HCT15 CRC cells (*Zheng et al., 2000*; *Figure 6—figure supplement 3*), ZNF304, KAP1, SETDB1, and DNMT1 were associated with the promoters of the nine CIMP marker genes (*Figure 3—figure supplement 2*, *Figure 6—figure supplement 4*), whose expression was derepressed by shRNA-mediated knockdown of ZNF304 or KRAS (*Figures 2A and 3A*, *Figure 6—figure supplement 5*).

The PAT-ChIP results of *Figures 3I and 6D* show that ZNF304 was substantially enriched at the promoters of the nine CIMP marker genes in KRAS-positive human CRC tumors relative to matched normal colon. Bisulfite sequencing analysis of representative CIMP marker genes confirmed promoter hypermethylation in all KRAS-positive CRC tumor samples (*Figure 6—figure supplement 6*). These results and those described above reveal a specific pathway that mediates CIMP in KRAS-positive CRCs that is summarized in *Figure 6E* and discussed below.

## ZNF304 also directs transcriptional silencing of *INK4-ARF* in hESCs

As described above, the *INK4-ARF* locus is also transcriptionally silenced in undifferentiated hESCs and becomes poised for expression following differentiation (*Sherr, 2012*). We therefore analyzed a possible role for ZNF304 and its corepressors in H9 cells, a well-characterized hESC line. The immunoblot results of *Figure 7A* show that in undifferentiated H9 hESCs, ZNF304 was present at high levels, which markedly decreased following differentiation by retinoic acid treatment. Moreover, in undifferentiated H9 hESCs ZNF304, KAP1, SETDB1, and DNMT1 were substantially enriched at the *p14^ARF*, *p15^INK4B*, and *p16^INK4A* promoters (*Figure 7B*), which was largely lost following differentiation. Finally, knockdown of ZNF304 in undifferentiated H9 hESCs (*Figure 7—figure supplement 1*) resulted in decreased

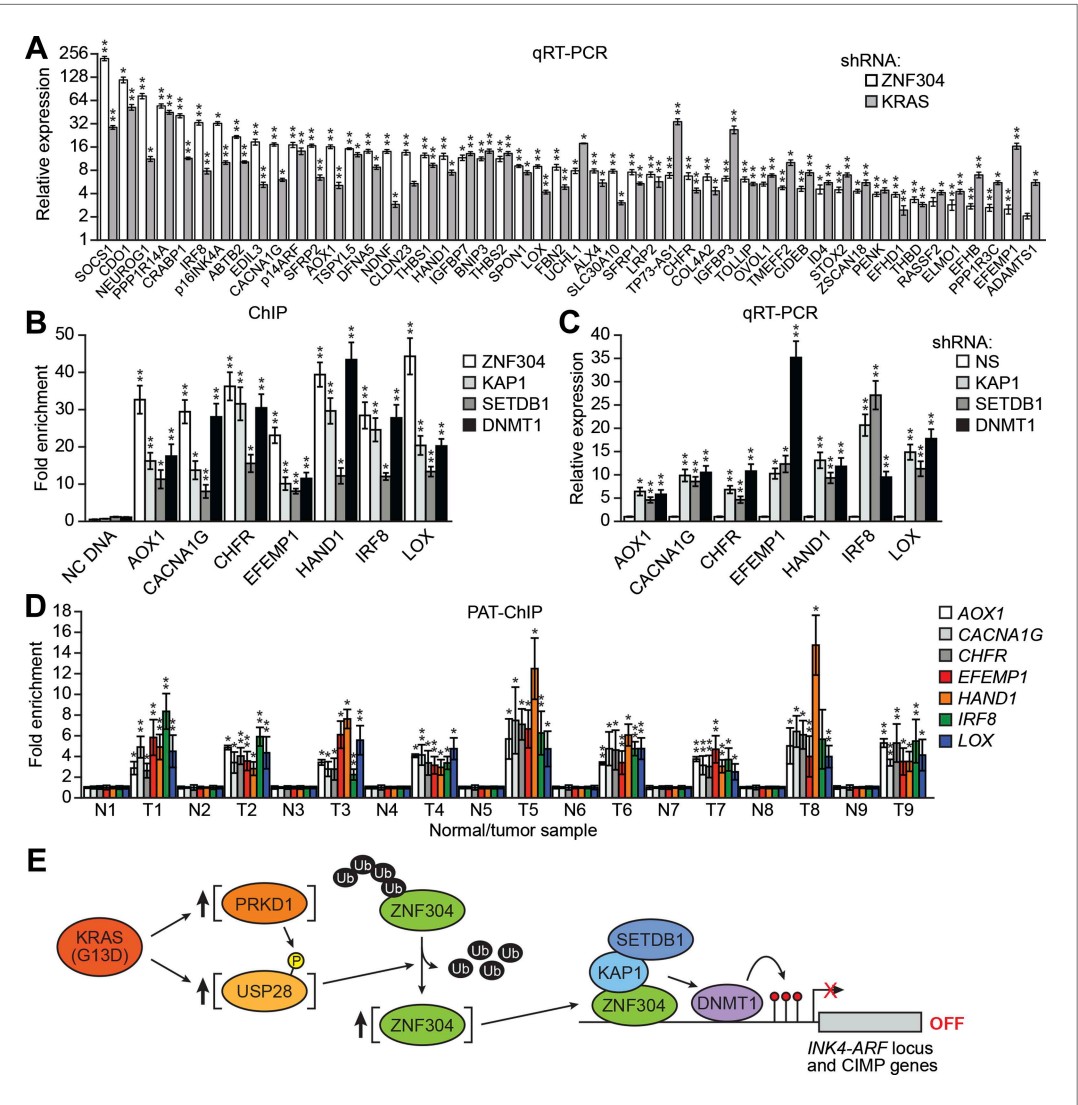

**Figure 6**. The ZNF304 corepressor complex mediates CIMP in KRAS-positive CRCs. (**A**) qRT-PCR analysis monitoring expression of CIMP marker genes in DLD-1 cells expressing a ZNF304 or KRAS shRNA. The results were normalized to that obtained with the NS control, which was set to 1. (**B**) ChIP analysis monitoring binding of ZNF304, KAP1, SETDB1, and DNMT1 to CIMP promoters or an irrelevant DNA region (NC). (**C**) qRT-PCR analysis monitoring expression of CIMP marker genes in DLD-1 cells expressing a NS, KAP1, SETDB1, or DNMT1 shRNA. (**D**) PAT-ChIP analysis monitoring binding of ZNF304 to CIMP promoters in matched adjacent normal (N) and KRAS-positive CRC human tumor (T) samples. Data are represented as mean ± SD. *p≤0.05, **p≤0.01. (**E**) Model for ZNF304-corepressor-mediated transcriptional silencing of *INK4-ARF* and CIMP marker genes in CRCs. Experiments validating the role of ZNF304 and corepressors in silencing of CIMP genes in KRAS-positive CRC cell lines, and experiments showing that the promoters of CIMP genes are hypermethylated in KRAS-positive CRC tumor samples, are presented in *Figure 6—figure supplements 1–6*.

The following figure supplements are available for figure 6:

**Figure supplement 1**. Knockdown of KRAS or ZNF304 derepresses expression of *VIM* and *SEPT9* in DLD-1 cells.

**Figure supplement 2**. Knockdown of KRAS or ZNF304 decreases promoter hypermethylation of representative CIMP genes in DLD-1 cells.

**Figure supplement 3**. Confirmation of CIMP in other KRAS-positive human CRC cell lines.

*Figure 6. Continued on next page*

*Figure 6. Continued*

**Figure supplement 4**. ZNF304 and its corepressors are associated with the promoters of representative CIMP genes in other KRAS-positive human CRC cell lines.

**Figure supplement 5**. Knockdown of KRAS or ZNF304 derepresses expression of representative CIMP genes in other KRAS-positive human CRC cell lines.

**Figure supplement 6**. Confirmation of CIMP in KRAS-positive human CRC tumor samples.

association of KAP1, SETDB1, and DNMT1 with the $p14^{ARF}$, $p15^{INK4B}$, and $p16^{INK4A}$ promoters (**Figure 7C**) and increased expression of $p14^{ARF}$, $p15^{INK4B}$, and $p16^{INK4A}$ (**Figure 7D**). Collectively, these results indicate that ZNF304 and its corepressors are also responsible for transcriptional silencing of *INK4-ARF* in undifferentiated hESCs.

As described above, in DLD-1 and other KRAS-positive CRC cell lines and tumors, the *INK4-ARF* locus is extensively hypermethylated. However, the bisulfite sequencing results of **Figure 7E** show that unlike DLD-1 cells, in H9 hESCs the $p14^{ARF}$ and $p16^{INK4A}$ promoters were not hypermethylated. We therefore sought to investigate how ZNF304 promoted *INK4-ARF* silencing in H9 hESCs in the absence of DNA hypermethylation. As mentioned above, previous studies have described a role for Polycomb repressive complexes (PRCs) in silencing of *INK4-ARF* in several non-malignant cell types including fibroblasts, mouse ESCs, and adult stem cells (*Jacobs et al., 1999*; *Molofsky et al., 2005*; *Bracken et al., 2007*; *Li et al., 2009*).

We therefore investigated the possible role of PRCs in *INK4-ARF* silencing in H9 hESCs and, for comparison, in KRAS-positive DLD-1 CRC cells. We first analyzed the *INK4-ARF* locus for the presence of the inhibitory histone mark, H3K27me3, which is catalyzed by the PRC2 subunit EZH2. The ChIP results of **Figure 7F** show substantial enrichment of H3K27me3 at the $p14^{ARF}$, $p15^{INK4B}$, and $p16^{INK4A}$ promoters in H9 hESCs, whereas in DLD-1 cells the level of H3K27me3 was substantially lower. By contrast, the level of histone H3 lysine 9 trimethylation (H3K9me3) at the $p14^{ARF}$, $p15^{INK4B}$, and $p16^{INK4A}$ promoters was much higher in DLD-1 cells than in H9 hESCs. As expected, there was no significant enrichment of histone H3 lysine 4 trimethylation (H3K4me3), a mark associated with transcription activity, in either DLD-1 cells or H9 hESCs.

The results described above indicated that although *INK4-ARF* is silenced in both KRAS-positive CRC cells and H9 hESCs, the inhibitory chromatin marks differ. To determine whether the differential inhibitory marks resulted from selective recruitment of repressive cofactors, we performed ChIP experiments. The results of **Figure 7G** show that in both H9 hESCs and DLD-1 cells, the PRC2 subunit EZH2 and the PRC1 subunit BMI1 were associated with the $p14^{ARF}$, $p15^{INK4B}$, and $p16^{INK4A}$ promoters. Thus, although the inhibitory marks on *INK4-ARF* differ in H9 hESCs and DLD-1 cells, the same set of repressive cofactors is present.

To determine whether ZNF304 was responsible for recruitment of PRC1 and PRC2, we performed ChIP experiments following depletion of ZNF304. **Figure 7H** shows that in both H9 hESCs and DLD-1 cells, knockdown of ZNF304 decreased binding of EZH2 and BMI1 to the $p14^{ARF}$, $p15^{INK4B}$, and $p16^{INK4A}$ promoters, which, as expected, was accompanied by a loss of H3K27me3 (**Figure 7I**). Also, as expected, in both H9 hESCs and DLD-1 cells, the knockdown of ZNF304 resulted in decreased H3K9me3 and increased H3K4me3 at the $p14^{ARF}$, $p15^{INK4B}$, and $p16^{INK4A}$ promoters (**Figure 7I**).

The differential levels of H3K27me3 on *INK4-ARF* described above raised the possibility that PRCs might have a more important role in silencing of $p14^{ARF}$, $p15^{INK4B}$, and $p16^{INK4A}$ in H9 hESCs compared to DLD-1 cells. Consistent with this idea, the knockdown of EZH2 or BMI1 (**Figure 7—figure supplement 2A,B**) increased expression of $p14^{ARF}$, $p15^{INK4B}$, and $p16^{INK4A}$ in H9 hESCs but not in DLD-1 cells (**Figure 7J**, **Figure 7—figure supplement 2C**).

## Discussion

It is well established that in many cancers specific genes affecting cellular growth control are hypermethylated and transcriptionally silenced (*Baylin, 2005*; *Esteller, 2006*). However, the mechanistic basis of epigenetic silencing is not understood. According to one model, an epigenetic event, such as hypermethylation of a CpG-rich promoter region of a TSG, occurs randomly due, for example, to loss

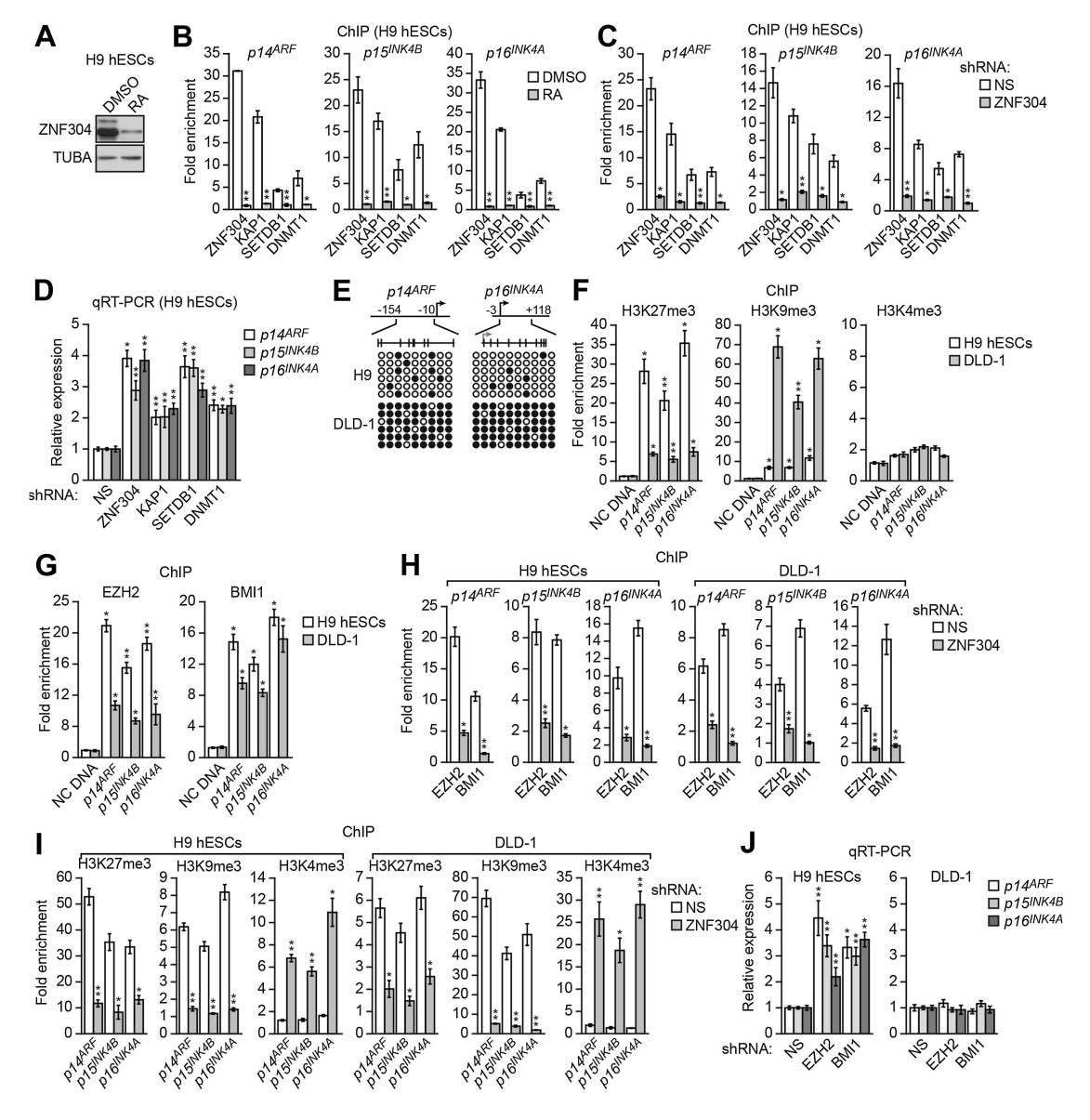

**Figure 7**. ZNF304 also directs transcriptional silencing of *INK4-ARF* in hESCs. (**A**) Immunoblot analysis showing ZNF304 levels in undifferentiated (DMSO) or retinoic acid (RA)-treated hESCs. (**B** and **C**) ChIP analysis monitoring binding of ZNF304, KAP1, SETDB1, and DNMT1 to *INK4-ARF* in undifferentiated or RA-treated hESCs (**B**) or in hESCs expressing a NS or ZNF304 shRNA (**C**). (**D**) qRT-PCR analysis monitoring *INK4-ARF* expression in hESCs expressing a NS, ZNF304, KAP1, SETDB1, or DNMT1 shRNA. (**E**) Bisulfite sequencing analysis of the *p14^ARF* and *p16^INK4A* promoters in H9 hESCs and DLD-1 cells. (**F** and **G**) ChIP analysis monitoring enrichment of H3K27me3, H3K9me3, and H3K4me3 (**F**) and EZH2 and BMI1 (**G**) at *INK4-ARF* or an irrelevant DNA region (NC) in H9 hESCs and DLD-1 cells. (**H** and **I**) ChIP analysis monitoring binding of EZH2 and BMI1 (**H**) and H3K27me3, H3K9me3 and H3K4me3 (**I**) at *INK4-ARF* in H9 hESCs and DLD-1 cells expressing a NS or ZNF304 shRNA. (**J**) qRT-PCR analysis monitoring *INK4-ARF* expression in H9 hESCs and DLD-1 cells expressing an NS, EZH2, or BMI1 shRNA. Data are represented as mean ± SD. *p≤0.05, **p≤0.01. Control experiments related to *Figure 7* are shown in *Figure 7—figure supplements 1,2*.

The following figure supplements are available for figure 7:

**Figure supplement 1**. Knockdown efficiencies in H9 hESCs.

**Figure supplement 2**. Assessment of the role of EZH2 and BMI1 in *INK4-ARF* transcriptional silencing in H9 hESCs and DLD-1 cells.

of fidelity or mutation of an epigenetic enzyme. The hypermethylation results in silencing of the TSG, which confers a selectable growth advantage (reviewed in *Hassler and Egger, 2012*). In a second model, transcriptional silencing occurs through a specific pathway, comprising a defined set of components, initiated by an oncoprotein.

## A ZNF304-corepressor complex mediates *INK4-ARF* silencing in KRAS-positive CRCs

In this report, we have identified a specific pathway that mediates CIMP in KRAS-positive CRCs (*Figure 6E*). The pathway is initiated on DNA by binding of the transcriptional repressor, ZNF304, which recruits a corepressor complex that includes SETDB1, KAP1 and DNMT1, leading to promoter hypermethylation and transcriptional silencing. Activated KRAS regulates the pathway by maintaining high levels of ZNF304, which drives DNA binding. The basis by which activated KRAS increases ZNF304 levels is transcriptional upregulation of PRKD1, a serine/threonine kinase, and USP28, a deubiquitinase, two of the factors that were isolated in our primary RNAi screen. We further showed that PRKD1 phosphorylates USP28, which interacts with and stabilizes ZNF304 from proteolytic degradation.

ZNF304 is a member of large family of transcription factors. About two-thirds of the approximately 1500 transcription factors encoded by mammalian genomes contain C2H2 zinc-fingers and more than half of these harbor an N-terminal KRAB transcriptional repression domain (KRAB-ZFP proteins; reviewed in *Lupo et al., 2013*). In a previous study, we have found that silencing of the *Fas* tumor suppressor in RAS-transformed NIH 3T3 cells requires ZFP354B (*Gazin et al., 2007*), which like ZNF304 is a KRAB-ZFP protein. Moreover, like ZNF304, ZFP354B levels are increased by activated RAS. These findings suggest that KRAB-ZFP transcription factors may have a widespread role in transcriptional silencing of TSGs in cancer cells.

## Implications for the role of KRAS in CIMP and tumorigenicity

Approximately 70% of CRCs containing activated KRAS are CIMP-positive. However, whether activated KRAS is merely associated with or is directly responsible for CIMP remained to be determined. In this study, we have shown that activated KRAS directs a pathway that silences *INK4A-ARF* and a large number of other genes characteristic of CIMP-positive CRCs. Our results show how a single oncoprotein-directed pathway can silence multiple, unrelated genes.

Although we have shown that in CRC cells KRAS directs and is required to maintain transcriptional silencing of *INK4A-ARF*, in primary cells oncogenic signals (such as activated KRAS) induce transcription of *INK4A-ARF*, which leads to p53 and Rb pathway activation and ultimately growth arrest (*Sherr, 2012*). Thus, transformation of a primary to a cancer cell involves a switch converting KRAS from an activator to a repressor of *INK4-ARF*. Consistent with this idea, we found, as expected, that expression of activated KRAS(G12V) in non-transformed WI-38 fibroblasts transcriptionally activated $p14^{ARF}$, $p15^{INK4B}$, and $p16^{INK4}$ (*Figure 8A*). However, in contrast to the results in KRAS-positive CRC cell lines, in WI-38 cells KRAS failed to increase ZNF304 protein levels (*Figure 8B*) or significantly stimulate *USP28* and *PRKD1* transcription (*Figure 8C*), explaining at least in part why KRAS expression does not result in *INK4-ARF* silencing. These results in WI-38 cells reinforce the pivotal role of ZNF304 and the KRAS-directed pathway we describe in mediating *INK4-ARF* silencing in KRAS-positive CRCs. We speculate that linking *INK4-ARF* silencing directly to KRAS may ensure that the locus is not reactivated in CRC tumor cells.

Collectively, our results indicate that the ZNF304 pathway mediates transcriptional silencing of $p14^{ARF}$, $p15^{INK4B}$, and $p16^{INK4A}$ and other TSGs thereby facilitating RAS-driven tumorigenicity. Thus, in addition to its well-established role in promoting cellular proliferation and preventing apoptosis through downstream signaling pathways (reviewed in *Karnoub and Weinberg, 2008*), RAS induces 'secondary' oncogenic events by inactivating TSGs through transcriptional silencing. This additional activity may explain, at least in part, why RAS is such a potent oncoprotein and why activating RAS mutations are found at high frequency in human tumors.

Finally, the continual requirement of the components of the RAS-directed ZNF304 pathway to maintain silencing has therapeutic implications. The approval of DNA demethylating agents and histone deacetylase inhibitors for treatment of lymphoma patients clearly established the potential to reverse tumor-specific epigenetic alterations (*Mann et al., 2007*; *Kelly et al., 2010*). However, DNA methyltransferase and histone deacetylase inhibitors broadly and non-selectively interfere with silencing. More efficacious therapeutics may be obtained by selectively inhibiting the silencing pathway initiated

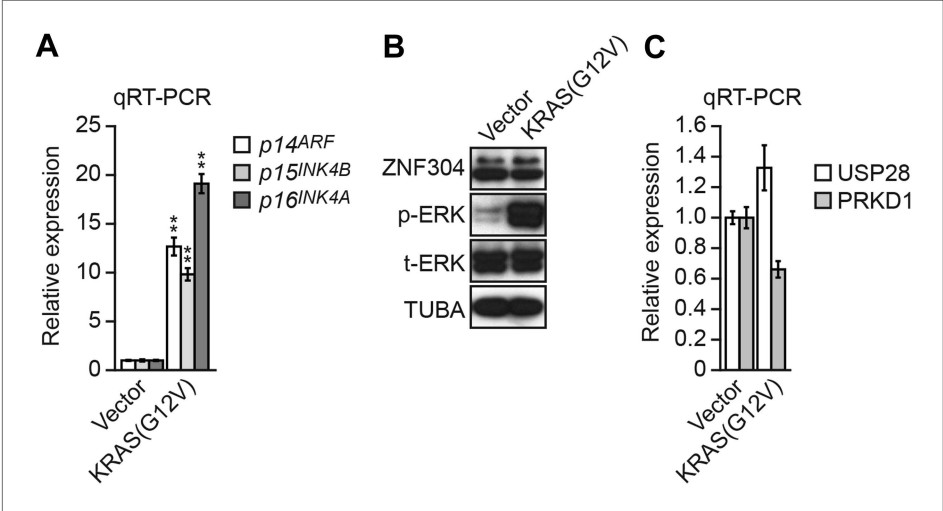

**Figure 8**. Expression of activated KRAS in non-transformed WI-38 fibroblasts increases expression of *INK4-ARF* and does not activate the ZNF304 pathway. (**A**) qRT-PCR analysis monitoring *INK4-ARF* expression in WI-38 cells expressing vector or KRAS(G12V). The results were normalized to the expression level obtained in vector-expressing cells, which was set to 1. (**B**) Immunoblot analysis showing levels of ZNF304, phosphorylated ERK (p-ERK) and total ERK (t-ERK) in WI-38 cells expressing vector or KRAS(G12V). (**C**) qRT-PCR analysis monitoring expression of *USP28* and *PRKD1* in WI-38 cells expressing vector or KRAS(G12V). The results were normalized to the expression level obtained in vector-expressing cells, which was set to 1. Data are represented as mean ± SD. *p≤0.05, **p≤0.01.

by the relevant oncoprotein. Thus, factors such as USP28 and PRKD1 may provide attractive anti-cancer targets.

## Transcriptional silencing of *INK4-ARF* in hESCs is directed by ZNF304

Unexpectedly, we have found that silencing of *INK4A-ARF* in undifferentiated hESCs is also mediated by ZNF304-directed recruitment of repressive cofactors. We note that the fold-increase of $p14^{ARF}$, $p15^{INK4B}$, and $p16^{INK4A}$ expression following knockdown of ZNF304 or repressive cofactors in H9 hESCs was less than that observed in KRAS-positive CRC cell lines, but is entirely consistent with results of other studies analyzing the role of *INK4-ARF* repressors in non-malignant cells (e.g., *Bracken et al., 2007*). This lack of robust re-expression of *INK4-ARF* in H9 hESCs and other non-malignant cells is presumably due to the absence of oncogenic signals that would strongly stimulate transcription of *INK4-ARF* genes.

ZNF304 is present at high levels in hESCs despite the absence of activated RAS. Interestingly, in ESCs KRAB-ZFPs, and their corepressor KAP1, have been shown to play an important role in silencing of endogenous retroviruses (*Rowe et al., 2013*). Thus, there may be some general mechanism for activation of KRAB-ZFPs in ESCs. Our finding that ZNF304 levels are elevated in both KRAS-positive CRC cells and hESCs is consistent with studies reporting a variety of similarities between cancer cells and stem cells (reviewed in *Kim and Orkin, 2011*).

In non-malignant differentiated cells *INK4-ARF* is reversibly silenced through association with PRCs. Although ZNF304 mediates recruitment of PRCs to *INK4-ARF* in undifferentiated H9 hESCS, following differentiation ZNF304 levels dramatically decline and there is no longer significant binding of ZNF304 at the $p14^{ARF}$, $p15^{INK4B}$, and $p16^{INK4A}$ promoters. Several other transcription factors such as TWIST1, ZFP277, HLX1, HOXA9 and the long non-coding RNA ANRIL have been reported to recruit PRCs and repress *INK4-ARF* in various differentiated non-malignant cells that presumably lack high levels of ZNF304 (*Negishi et al., 2010*; *Yang et al., 2010*; *Yap et al., 2010*; *Martin et al., 2013*; *Ribeiro et al., 2013*).

In ESCs, many developmentally important genes are reversibly silenced by PRCs and contain the repressive H3K27me3 mark. In fact, in ESCs H3K27me3 is a more predominant silencing mechanism than DNA hypermethylation, which is believed to be because H3K27me3 is more readily reversible. Several studies have identified subgroups of genes in which the H3K27me3 present in ESCs is replaced

by abnormal DNA hypermethylation in cancer cells. This phenomenon, which has been referred to as 'epigenetic switching', results in the permanent silencing of key regulatory genes that may contribute to cell proliferation and tumorigenesis (reviewed in *Sharma et al., 2010*).

This differential repression pattern is exactly what we found for *INK4-ARF* in hESCs and KRAS-positive CRC cells. One model to explain the two repression patterns is differential recruitment of corepressors, such as PRCs or DNMTs, in non-malignant and cancer cells. However, we found that in hESCs and KRAS-positive CRC cells the same set of corepressors are recruited to *INK4-ARF*. Moreover, corepressor recruitment is dependent upon the same sequence-specific DNA-binding protein, ZNF304. Our results suggest a model involving differential enzymatic activity of a common, promoter-bound corepressor complex resulting in predominantly DNA hypermethylation, in KRAS-positive CRC cells, or predominantly H3K27me3, in hESCs.

# Materials and methods

## Cell lines and culture

DLD-1, HCT15, and HCT116 cells were obtained from ATCC (Manassas, VA) and grown as recommended by the supplier. H9 hESCs were grown in mTeSR1 media (STEMCELL Technologies, Vancouver, Canada) under feeder-free conditions on plates coated with Matrigel (BD Biosciences, San Jose, CA). DLD-1 cells were treated with 10 µM manumycin A (Calbiochem, Darmstadt, Germany) for 24 hr, 20 µM LY294002 (Calbiochem) for 24 hr, 10 µM PI-103 (Cayman Chemical) for 24 hr, 0–10 µM MG-132 (Cayman Chemical, Ann Arbor, MI) for 4 hr, or 10 µM CRT0066101 (Cancer Research Technology, London, UK) for 24 hr. To induce differentiation, H9 hESCs were treated with 10 µM retinoic acid (Sigma–Aldrich, St. Louis, MO) for 3 (*Figure 7A*) or 4 (*Figure 7B*) days.

WI-38 cells were obtained from ATCC and cultured as recommended by the supplier. To produce retroviruses, HEK293T cells were tranfected with pBabe K-Ras12V (12544; Addgeneplasmid; *Khosravi-Far et al., 1996*) or empty vector (pBABE-puro; 1764; Addgene plasmid). WI-38 cells were seeded with the KRAS or empty vector retrovirus and 3 days later selected with puromycin. The cells were harvested for total RNA for qRT-PCR analysis on day 4, or for total protein for immunoblot analysis on day 7.

## Reporter construct cloning and validation

To construct the *p14^ARF^-Blast^R^* reporter, 3.98 kb of the *p14^ARF^* promoter was PCR amplified from a BAC using primers engineered with *Bgl*II and *Sal*I restriction sites, and cloned into a derivative of pDsRed2-N1 (Clontech, Mountainview, CA) in which the CMV promoter had been excised, the *Blast^R^* gene (PCR amplified from pEF6/V5-HisB; Invitrogen, Grand Island, NY) had been inserted in-frame with DsRed2, and the TK gene (PCR amplified from the HSV-1-TK gene; Addgene) had been inserted in-frame with DsRed2-*Blast^R^*. The plasmid was linearized and stably transfected into DLD-1 cells using an Amaxa nucleofactor. Immediately after nucleofection, the cells were placed in complete growth medium for 72 hr. Viable cells were allowed to grow into colonies and then selected with 500 µg/ml G418 (Calbiochem). Surviving colonies were individually isolated, expanded, and tested for blasticidin sensitivity; to minimize non-specific survival in the shRNA screen, clones with the lowest resistance to blasticiden, indicating complete reporter silencing, were chosen for further characterization. Clones were treated with 10 µM 5-aza-2′-deoxycytodine (Calbiochem) every 24 hr for 72 hr. After 24 hr treatment, 0, 5, or 10 µM blasticidin (Sigma-Aldrich) was added for 6 days, and cells were fixed and stained with 0.1% crystal violet to assess viability. Treatment with 5-aza-2′-deoxycytodine and subsequent challenge with blasticidin was used to identify a clone with robust survival when treated with both drugs.

## shRNA screen

The human shRNA^mir^ library (release 1.20; Open Biosystems/Thermo Scientific, Pittsburgh, PA) was obtained through the UMass Medical School RNAi Core facility (Worcester, MA). Retroviral pools were generated and used to transduce DLD-1/*p14^ARF^-Blast^R^* cells as previously described (*Gazin et al., 2007*). The cells were selected with puromycin (4 µg/ml) for 3 days, and the puromycin-resistant population was challenged with blasticidin (10 µg/ml) for 14 days. The cells that bypassed the basticidin challenge formed colonies that were isolated and individually expanded, and shRNAs were identified by sequence analysis as previously described (*Gazin et al., 2007*). Individual knockdown cell lines were generated by stable transduction of 1 × 10^5 cells with a single shRNA (*Supplementary file 1*) followed by puromycin selection.

## qRT-PCR

Total RNA was isolated and reverse transcription was performed as described (*Gazin et al., 2007*), followed by qRT-PCR using Power SYBR Green PCR Master Mix (Applied Biosystems, Grand Island, NY). *GAPDH* was used as an internal reference gene for normalization. See *Supplementary file 2* for primer sequences.

## Immunoblot analysis

Cell extracts were prepared by lysis in Laemmli buffer in the presence of protease inhibitor cocktail (Roche, Indianapolis, IN). The ZNF304 antibody was generated (by 21st Century Biochemicals, Marlboro, MA) against a peptide corresponding to amino acids GFWCEAEHEAPSEQSV. The following commercial antibodies were used: p14ARF (Cell Signaling Technology, Danvers, MA), p15INK4B (Abcam, Cambridge, MA), p16INK4A (Cell Signaling Technology), USP28 (Bethyl Laboratories, Montgomery, TX), PRKD1 (Cell Signaling Technology), phospho-ERK1/2 and total ERK1/2 (both from Cell Signaling Technology). The α-tubulin (TUBA) antibody was generated in-house.

## ChIP assays

ChIP assays were performed as previously described (*Gazin et al., 2007*) using the following antibodies: ZNF304 (described above), KAP1 (Bethyl Laboratories), SETDB1 (Millipore, Billerica, MA), DNMT1, DNMT3A, and DNMT3B (all from Imgenex, San Deigo, CA), cJUN (Millipore), H3K27me3 (Cell Signaling Technology), H3K9me3 (Millipore), H3K4me3 (Abcam), EZH2 (Millipore) and BMI1 (Abcam). The CDX1 antibody (*Chan et al., 2009*) was kindly provided by Walter Bodmer (University of Oxford, UK). ChIP products were analyzed by qRT-PCR (see *Supplementary file 1* for primers). Samples were quantified as percentage of input, and then normalized to an irrelevant region in the genome (~3.2 kb upstream from the transcription start site of *GCLC*). Fold enrichment was calculated by setting the IgG control IP sample to a value of 1.

## Bisulfite sequencing

Bisulfite modification was carried out using an EpiTect Bisulfite Kit (QIAGEN, Germantown, MD) followed by assay kits from EpigenDX (Hopkinton, MA) or nested PCR primers. Multiple independent clones were sequenced from each PCR product within each cell line (see *Supplementary file 2* for primer sequences), of which six representative clones are displayed.

## Tumor formation assays

DLD-1 cells ($2 \times 10^6$) expressing either a NS, ZNF304 or DNMT1 shRNA were suspended in 100 μl of serum-free RPMI and injected subcutaneously into the right flank of athymic BALB/c (nu/nu) mice (Taconic) (n = 3 mice per shRNA). Tumor dimensions were measured every 7 days for 4 weeks and tumor volume was calculated using the formula $\pi/6 \times (\text{length}) \times (\text{width})^2$. All experiments were performed in accordance with the Institutional Animal Care and Use Committee (IACUC) guidelines.

## PAT-ChIP assay

This study was approved by the institutional review board at the University of Massachusetts Medical School (UMMS). Archived specimens (2010–2012) with sufficient tissue for analysis were obtained from the Department of Pathology at UMMS, and the CRC diagnosis was made by a UMMS pathologist. KRAS mutational analysis was performed by the UMass Memorial Laboratory of Diagnostic Molecular Oncology (Worcester, MA). Formalin-fixed paraffin embedded tissue sections of matched adjacent normal colon and tumor samples isolated from individuals with invasive or metastatic KRAS-positive CRC were de-paraffinized in Histolemon-Erba RS solution (Carlo Erba Reagents, France) four times for 10 min at room temperature. The tissue was then resuspended in 100% ethanol, incubated for 10 min at room temperature, spun down and resuspended in 95% ethanol. The washing/resuspension steps were repeated, gradually increasing the percentage of water (to achieve 70%, 50%, 20%, 0% ethanol) to rehydrate the tissue. The resulting material was then processed as previously described (*Fanelli et al., 2011*).

## Co-immunoprecipitation assays

DLD-1 cell lysate was immunopreciptated with a ZNF304, USP28, PRKD1 or control (IgG) antibody, and the immunoprecipitate was analyzed for ZNF304, USP28 or PRKD1 by immunoblotting. Input lanes represent 10% of immunoprecipitated lanes.

## HA-ubiquitin pull-down assays

ZNF304 was PCR amplified from a BAC using primers engineered with HindIII and BamH1 sites and cloned into p3XFLAG-myc-CMV-26 (Sigma-Aldrich) to generate p3XFLAG-ZNF304. p3XFLAG-USP28 and p3XFLAG-USP28(C171A) (*Popov et al., 2007*) were kindly provided by Stephen Elledge (Harvard Medical School, Boston, MA); p3XFLAG-USP28(S899A) was generated by patch PCR using p3XFLAG-USP28 as a template. 293T cells ($2 \times 10^6$) were plated on 10-cm dishes and transfected with 1 µg p3XFLAG-ZNF304, 1 µg p3XFLAG-USP28 (wild-type or mutant), 1 µg pcDNA3.1-HA-Ubiquitin (Addgene), and 0.5 µg pmaxGFP (Lonza Biologics Inc., Hopkinton, MA) using Effectene reagent (QIAGEN). To ensure equivalent transfection efficiency, eGFP expression was monitored 48 hr later. Cells were harvested in NETN-150 buffer (20 mM Tris–HCl, pH 8.0, 150 mM NaCl, 1 mM EDTA and 0.05% NP-40) plus 1X protease inhibitor cocktail (Roche). Pull-downs were performed using an HA antibody (Cell Signaling Technology) and anti-rabbit Trublot beads (eBioscience, San Diego, CA). Beads were incubated with lysate for 18 hr, washed three times using NETN-150 buffer, and eluted in 2X sample buffer. Input samples were probed with a FLAG-M2 (Sigma-Aldrich) or TUBA antibody, and immunoprecipitated samples were probed with a ZNF304 antibody.

## In vitro kinase assay

Plasmids expressing His-tagged wild-type and kinase-dead PRKD1 proteins were constructed by digesting plamids HA.PKD (*Storz and Toker, 2003*) and HA.PKD.K/W (*Storz et al., 2003*) (10808 and 10809; Addgene plasmids), respectively, with BamHI and XhoI and ligating into pRSET A (Life Technologies, Grand Island, NY). To ensure activity of the purified wild-type protein, a 20-µl reaction was set up as follows: 1 µl $^{32}$P-γ-ATP (10 mCi), 1 µl 10 µM ATP, 0.2 mM microcystin, 4 µl 5X kinase buffer (23 mM MOPS, 11.5 mM β-glycerphosphate, 23 mM MgCl$_2$, 4.6 mM EGTA, 1.8 mM EDTA, 0.25 mM DTT [pH 7.0]), and 60 nM purifiedHis-PRKD1 diluted in 1X kinase buffer. Reactions were incubated for 30 min at 30°C and stopped using 2X Laemmli Sample Buffer. Autophosphorylation of the wild-type protein was confirmed by immunoblotting with a PRKD1-Ser916 antibody (Cell Signaling Technology). To monitor phosphorylation of USP28, the above reaction was carried out using 10 µM substrate (peptides corresponding to amino acids 543–557 [TCLQRWRSEIEQDIQ] or 892–906 [YSLFRKVSVYLLTGL] in USP28) diluted in 1X kinase buffer. Incorporation of the radiolabel into the peptide was monitored by autoradiography.

## *USP28* and *PRKD1* promoter analysis

The *USP28* and *PRKD1* promoters were analyzed using the TRANSFAC database (www.gene-regulation.com/pub/databases.html) to identify putative transcription factor binding sites.

## Statistics

All quantitative data were collected from experiments performed in at least triplicate, and expressed as mean ± standard deviation. Differences between groups were assayed using two-tailed student *t* test using Microsoft Excel. Significant differences were considered when $p < 0.05$; *$p \leq 0.05$, and **$p \leq 0.01$.

## Acknowledgements

We thank Stephen J Elledge, Marco Ballarini, and Walter Bodmer for providing reagents; the UMMS RNAi Core Facility for providing shRNA clones and libraries; and Sara Deibler for editorial assistance.

## Additional information

### Funding

| Funder | Grant reference number | Author |
| --- | --- | --- |
| National Institutes of Health | R01GM033977 | Michael R Green |
| Howard Hughes Medical Institute | 068101 | Michael R Green |

The funder had no role in study design, data collection and interpretation, or the decision to submit the work for publication.

## Author contributions

RWS, Conception and design, Acquisition of data, Analysis and interpretation of data, Drafting or revising the article; MF, Acquisition of data, Analysis and interpretation of data, Drafting or revising the article; SMP, Performed the in vitro kinase assays, Acquisition of data; LH, Provided the human CRC samples, performed the genotyping and characterized the samples, Acquisition of data, Contributed unpublished essential data or reagents; MRG, Conception and design, Analysis and interpretation of data, Drafting or revising the article

## Ethics

Human subjects: Informed consent and consent to publish were not necessary, as archival specimens were retrieved from existing material and provided as anonymous, de-identified specimens. This study was approved by the institutional review board at the University of Massachusetts Medical School (UMMS). Animal experimentation: All experiments were performed in accordance with the Institutional Animal Care and Use Committee (IACUC) guidelines at the University of Massachusetts Medical School (protocol A-2247).

## Additional files

### Supplementary files

• Supplementary file 1. List of shRNAs obtained from Open Biosystems/Thermo Scientific and synthesized siRNA sequences. For shRNAs, clone IDs are provided, except in instances in which the shRNA clones have been discontinued, in which case sequences are provided.

• Supplementary file 2. List of primers used for qRT-PCR, ChIP, PAT-ChIP and bisulfite sequencing.

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
