## [Decision Letter]

Thank you for sending your work entitled “A KRAS-directed transcriptional silencing pathway that mediates the CpG island methylator phenotype” for consideration at *eLife*. Your article has been favorably reviewed by a Senior editor (Jim Manley) and a member of our Board of Reviewing Editors (Kevin Struhl).

We are very pleased to inform you that your article has been accepted for publication: this is a superb paper that deserves to be published in *eLife*. The authors describe a detailed pathway in which activated KRAS ultimately leads to DNA methylation of the INK4-ARF and a number of other CIMP loci. The pathway was elucidated through a genome-wide RNAi screen and a large number of standard molecular biological experiments. It involves a DNA-binding transcription factor and its associated co–repressor complex, a DNA methylase, a deubiquitinase, a protein kinase, and the AP-1 factor June Moreover, the authors provide clear evidence that this pathway is relevant in human cancer by analyzing patient samples. Lastly, they also show that aspects of this pathway also occur in embryonic stem cells, providing a new link between such stem cells and the cancer state. Although the Green laboratory published a seminal paper showing that DNA methylation of tumor suppressor genes can arise from a detailed genetic pathway as opposed to previous belief that methylation occurs randomly (and then propagated), this paper uses a completely different system. As such, it is important for its own sake as well as providing a new example of the general principle, which I think is largely under-appreciated in the haze of cancer and chromatin work. The paper is extremely well written, and I have no specific comments other than to cite the work of Sonia Kohai.

[Editors’ note: no revisions were requested so there is not an accompanying author response.]